# Metaplastic Matrix-Producing Carcinoma and Apocrine Lobular Carcinoma In Situ Associated with Microglandular Adenosis: A Unique Case Report

**DOI:** 10.3390/diagnostics12061458

**Published:** 2022-06-13

**Authors:** Nektarios Koufopoulos, Dionysios Dimas, Foteini Antoniadou, Kyparissia Sitara, Dimitrios Balalis, Ioannis Boutas, Alina Roxana Gouloumis, Adamantia Kontogeorgi, Lubna Khaldi

**Affiliations:** 1Second Department of Pathology, Medical School, National and Kapodistrian University of Athens, Attikon University Hospital, 15772 Athens, Greece; alina.gmed@gmail.com; 2Breast Unit, Athens Medical Center, Psychiko Clinic, 11525 Athens, Greece; dionysis.dimas@gmail.com; 3Department of Pathology, “Alexandra” General Hospital, 11528 Athens, Greece; fenia.antoniadou@gmail.com; 4Department of Internal Medicine, “Elpis” General Hospital of Athens, 11522 Athens, Greece; kdsitara@gmail.com; 5Department of Surgical Oncology, Saint Savvas Anticancer Hospital of Athens, 11522 Athens, Greece; dbalalis@gmail.com; 6Breast Unit, Rea Maternity Hospital, P. Faliro, 17564 Athens, Greece; ioannis.boutas@gmail.com; 7Third Department of Obstetrics and Gynecology, Medical School, National and Kapodistrian University of Athens, Attikon University Hospital, 15772 Athens, Greece; ad.kontogewrgi@gmail.com; 8Department of Pathology, Saint Savvas Anticancer Hospital of Athens, 11522 Athens, Greece; lubna.khaldi@gmail.com

**Keywords:** microglandular adenosis, atypical microglandular adenosis, breast carcinoma, metaplastic carcinoma, matrix-producing carcinoma, lobular carcinoma in situ, pleomorphic lobular carcinoma in situ, HER-2 positive, triple negative

## Abstract

Microglandular adenosis is a non-lobulocentric haphazard proliferation of small round glands composed of a single layer of flat to cuboidal epithelial cells. The glandular structures lack a myoepithelial layer; however, they are surrounded by a basement membrane. Its clinical course is benign, when it is not associated with invasive carcinoma. In around 30% of cases, there is a gradual transition to atypical microglandular adenosis, carcinoma in situ, and invasive breast carcinoma of several different histologic subtypes, including an invasive carcinoma of no special type, metaplastic matrix-producing carcinoma, secretory carcinoma, metaplastic carcinoma with squamous differentiation, acinic cell carcinoma, spindle cell carcinoma, and adenoid cystic carcinoma. Recent molecular studies suggest that microglandular adenosis is a non-obligate precursor of triple-negative breast carcinomas. In this manuscript, we present a unique case of microglandular adenosis associated with metaplastic matrix-producing carcinoma and HER-2 neu oncoprotein positive pleomorphic lobular carcinoma in situ with apocrine differentiation in a 79-year-old patient.

## 1. Introduction

Microglandular adenosis (MGA) is a unique lesion in breast pathology. It consists of an invasive haphazard proliferation of small uniform round tubular structures with open lumens lined with a single layer of cuboidal epithelial cells. MGA lacks a myoepithelial layer, but a basement membrane surrounds it. An eosinophilic colloid-like secretion is found inside several of the open luminal spaces. Its correct diagnosis may be difficult since it simulates invasive carcinoma clinically, radiologically, and histologically [1,2,3]. Uncomplicated MGA has an indolent clinical course with no recurrence reported to date [4]. MGA is associated with invasive breast carcinoma in 27% of cases [5]. In these cases, there is a gradual transition from MGA to atypical microglandular adenosis (AMGA), carcinoma in situ (CIS), and invasive carcinoma [6]. The invasive carcinomas associated with MGA belong to several different histologic subtypes. (Invasive carcinoma of no special type, metaplastic matrix producing carcinoma (MMPC), secretory carcinoma, carcinoma with metaplastic squamous differentiation, acinic cell carcinoma, spindle cell carcinoma, and adenoid cystic carcinoma). MGA displays similar alterations to the invasive carcinoma on the molecular level, unlike uncomplicated MGA.

We present a unique case of MGA merging into an AMGA and metaplastic matrix-producing carcinoma (MMPC) and HER-2neu oncoprotein positive pleomorphic lobular carcinoma in situ (PLCIS) with apocrine differentiation. This case highlights the heterogeneous nature of MGA and the associated carcinomas.

## 2. Case Report

A 79-year-old female patient was admitted to our hospital due to a palpable lump on her right breast noticed on self-examination. Upon mammography, there was no significant abnormality. An ultrasound revealed an area of mild architectural distortion at the upper outer quadrant of the right breast with indistinct margins and a vertical orientation. Its maximum diameter was 9 mm and had an intermediate score on strain elastography (elastography index: 2.9). A breast MRI was performed, demonstrating an area of pathological contrast enhancement with dimensions 2.3 × 2.2 cm and irregular margins. There was a 1 cm tumor with suspicious characteristics (low-density mass with a peripheral ring structure enhancement with irregular borders) anterior to this lesion. At the medial border of the lesion, the MRI showed an area of linear pathological contrast uptake, with an anteroposterior diameter of 2 cm. A fine-needle aspiration biopsy was performed. The diagnosis was invasive carcinoma with no special type and an Elston–Ellis grade 3. A computed tomography of the upper and lower abdomen, positron emission tomography–computed tomography, and bone scintigraphy did not reveal metastatic disease. A mastectomy and a sentinel lymph node biopsy were performed.

On gross examination, the lesion was solid, with a hard consistency, was gray-white, and measured 45 mm in its greater dimension. On microscopic examination, the lesion consisted mainly of a haphazard non-lobulocentric proliferation composed of small round uniform tubules lined by a single layer of cuboidal epithelial cells with small nuclei and an amphophilic cytoplasm. A colloid-like eosinophilic luminal secretion was present in some tubules. This proliferation showed a transition to areas of greater architectural complexity and invasive carcinoma consisting of solid areas, small nests, tubular and ring-like structures, cords, and single cells, with an abrupt transition to a chondromyxoid matrix. The matrix was diffuse, accounting for 50–60% of the carcinomatous component. The tumor cells were pleomorphic, with enlarged nuclei and visible nucleoli, and without an intervening spindle cell component. The mitotic activity was high. An Elston–Ellis grade 3 was assigned. Necrosis and peripheral lymphocytic infiltrations were focally present. Angiolymphatic invasion was not identified. Located among them was an intraductal proliferation composed of moderate to high nuclear grade tumor cells with eosinophilic granular cytoplasm and central to eccentric hyperchromatic nuclei with occasional prominent nucleoli seemingly lacking cohesion (Figure 1). 

The tubules, as well as the invasive carcinoma component, were stained for S-100. The haphazardly arranged tubules were negative for myoepithelial markers P63, CK5/6, and smooth muscle actin with laminin highlighting the presence of a basement membrane. The intraductal proliferation showed negative E-cadherin and positive androgen receptors (AR) and AMACR, and P63 (peripherally on myoepithelial cells) staining. The HER2-neu oncoprotein was positive (score 3+) in the intraductal proliferation. The estrogen receptors (ER) and progesterone receptors (PR) were uniformly negative. The proliferative index Ki-67 stained 70% of the tumor cells in the invasive carcinoma (Figure 2).

The immunohistochemical features of all tumor components are presented in Table 1.

Our diagnosis was MMPC associated with MGA with the simultaneous presence of PLCIS with an apocrine differentiation based on the morphological and immunohistochemical findings. The maximum diameter of the invasive carcinoma was 18 mm. The sentinel lymph node biopsy was negative. The pathological TNM stage was T1c N0(sn) stage IA. The patient received six cycles of adjuvant therapy consisting of cyclophosphamide, methotrexate, and 5-Fluorouracil, and is alive without evidence of recurrence or metastasis 22 months after surgery.

## 3. Discussion

MGA is a rare proliferative glandular breast lesion mimicking carcinoma both histologically and clinically [6]. Its course is benign when devoid of invasive carcinoma. The exact nature and potential role in carcinogenesis acting as a precursor lesion are still controversial [7]. The direct transition of MGA to AMGA and invasive breast carcinoma and the shared immunohistochemical expression of these lesions (triple-negative and S-100 positive) suggest that, in at least some cases, MGA may act as a non-obligate precursor of triple-negative breast carcinomas [8].

Clinically, MGA may be present as a tumor-forming lesion or an incidental microscopic finding, while MGA-associated carcinomas (MGACA) are present as a palpable mass [9]. A specific mammographic pattern for MGA has not been described to date [4]. Uncomplicated MGA may be undetectable on mammography, while AMGA and MGACA present as infiltrative mass lesions occasionally associated with microcalcifications [10,11].

Histologically, the differential diagnosis of MGA includes AMGA, CIS, and tubular carcinoma. AMGA displays a greater architectural complexity and cytological atypia compared to MGA. The architectural complexity consists of interconnected budding glandular units with cribriform nests. Mitoses are usually present. Intraluminal eosinophilic secretions are diminished or absent [6]. CIS is characterized by a greater degree of cytologic atypia than AMGA but with the retention of the alveolar growth pattern.

In contrast to the round glands of MGA, those of tubular carcinoma are angulated and typically lack a basement membrane. Moreover, intraductal carcinoma is associated with some tubular carcinomas. Immunohistochemically, tubular carcinoma cells are ER+/PR+ [9].

Immunohistochemical and histochemical stains may aid in the differential diagnosis of the lesions described above. Stains for laminin and collagen IV will highlight the presence of a basement membrane in MGA, AMGA, and CIS while myoepithelial markers are negative. The presence of a basement membrane is characteristic of these lesions and helps in the distinguishing them from their mimics. In any apocrine lesion, the morphology should be supported by a certain immunophenotype (ER-, PR-, GCDFP-15+, androgen receptor+, and P504S+). In the case of LCIS, a lack of staining for E-cadherin is helpful in the differential diagnosis against other apocrine lesions.

Molecular studies support the hypothesis that MGA is a non-obligate precursor of triple-negative breast carcinoma [12,13]. Recently, a study has shown evidence suggesting the molecular progression of MGA to MMPC [14]. Guerrini et al. found identical TP53 mutations and similar patterns of gene copy number alterations in the MGA and/or AMGA and in the associated invasive component [15]. Pareja et al. detected highly recurrent TP53 mutations and occasional PIK3CA hotspot mutations. Radner et al. a found copy number gain on chromosome 2q and an epigenetic inactivation of GATA3 [16,17]. MGACAs belong to various histological subtypes displaying a triple-negative immunophenotype. Invasive carcinoma of no special type is the most common type of carcinoma associated with MGA [8]. Other variants of invasive carcinoma include MMPC, secretory carcinoma, carcinoma with metaplastic squamous differentiation, acinic cell carcinoma, spindle cell carcinoma, and adenoid cystic carcinoma [4,8]. MGA has also been associated with ductal carcinoma in situ, lobular carcinoma, and adenomyoepithelioma [4]. In the literature, only four cases of MGACA displayed a non-triple negative immunophenotype being ER+/PR+/HER-2 neu- (two cases), ER-/PR+/HER-2neu-, and ER-/PR-/HER-2neu+ [18,19,20]. To our knowledge, our case is the first in the English literature reporting an MGACA with the concomitant presence of two histologically distinct types of carcinomas. Moreover, the immunophenotype of these carcinomas is different (triple-negative versus HER-2 neu positive), suggesting that MGA is heterogeneous with a potential for simultaneous dual differentiation. This fact raises the question of whether MGA may display a simultaneous transition to a mixed-type carcinoma with a different immunophenotype. The clonal relationship of MGA to both carcinoma components would merit investigation at the molecular level, but we were unable to do so due to technical limitations.

MMPC is a rare subtype of metaplastic breast carcinoma that was initially described in 1988 by Wargotz et al. [21]. Despite its rarity, it shows an increased frequency in MGACAs. It is characterized by invasive carcinoma transitioning directly to a cartilaginous and/or osseous stromal matrix lacking an intervening spindle cell component [22]. Its differential diagnosis includes invasive breast carcinoma with a large central acellular zone, spindle cell carcinoma, primary or secondary chondrosarcoma, malignant phyllodes tumor with heterologous (chondrosarcomatous) differentiation, carcinoma ex-pleomorphic adenoma of the breast, and invasive lobular carcinoma with extracellular mucin production [6,23,24,25,26]. An inspection of the histological detail paired with the appropriate immunohistochemical stains will usually resolve diagnostic dilemmas [22]. The most frequent gene mutation in all the histological subtypes of metaplastic carcinomas is TP53, followed in frequency by PIK3CA, TERT, KTM2D, PIK3R1, PTEN, RB1, NF1, HRAS, and ARID1A [27]. The most frequent copy number variations include the amplification of MYC followed by EGFR and CCND3 and the deletion of CDKN2A/CDKN2B, PTEN, and RB1 [27]. Metaplastic carcinomas with a mesenchymal differentiation have frequent TP53 and PIK3CA mutations while lacking TERT promoter mutations [27].

PLCIS consists of a lobulocentric proliferation of high nuclear grade pleomorphic cells filling and distending lobules. Neoplastic cells have slightly granular or foamy eosinophilic cytoplasm and lack cohesion. Intracytoplasmic vacuoles can be seen frequently. In some cases, an apocrine differentiation can be observed [28]. A genetic analysis of PLCIS shows a clonal relationship to classic lobular carcinoma in situ, invasive lobular carcinoma, and frequent ERBB2/ERBB3 alterations [29].

The prognosis of MGACAs is controversial in the literature. Some authors report a better prognosis [9], while others report a variable prognosis ranging from favorable to unfavorable [8]. We can assume that the better prognosis reported, at least in some cases, is the result of the presence of MGA, which allows for the detection of carcinomas at an earlier stage.

Regarding treatment, MGA management consists of a complete excision. Cases with AMGA should be treated with a wide excision with clear margins and a careful follow-up. For patients with MGACAs, the guidelines for invasive breast carcinomas are followed with treatment consisting of breast-conserving surgery or mastectomy, combined with adjuvant therapy when necessary [9]. In our case, the patient was more than 70 years-old with a triple-negative breast carcinoma (TNBC). The multidisciplinary tumor board decided to administer an adjuvant systemic therapy which remains the backbone of TNBC treatment. Large trials and prospective cohorts suggest that in hormone-receptor negative breast cancer, women derive significant benefits from chemotherapy in terms of disease-specific and overall survival [30,31,32].

In older patients (>70 years old), as in our case, due to comorbidities and a reduced life expectancy, decisions about systemic treatment need to balance the potential benefit versus the drug-related toxicity. In fact, older patients more frequently experience adverse effects [33].

Our 79-year-old patient was, at the time of diagnosis, functionally independent, without any severe comorbidities. For this type of older but “fit” [34,35] patient, there is strong and growing evidence that there is a benefit from adjuvant chemotherapy that is likely to be the same as that of their younger counterparts [31,32,36].

The EUSOMA guidelines for older patients with breast cancer suggest that the decision on adjuvant treatment should not be based on chronological age alone [37]. Older and fit patients are eligible for systemic treatment, and adults with ER-negative tumors have the largest benefit from adjuvant chemotherapy. The EUSOMA and NCCN guidelines suggest adjuvant chemotherapy for patients with triple negative breast cancer tumors >1 cm, irrespective of nodal status [37,38].

Six cycles of CMF comprise a validated regimen in older patients that has a manageable and well-known toxicity profile based on more than thirty-five years of data. On the other hand, anthracycline and taxane based regimens pose a greater risk of haematological toxicity, cardiotoxicity (up to 38%), neurotoxicity, hair loss, falls, and hospitalizations in these patients. A clinical trial demonstrated a worse quality of life when comparing docetaxel versus CMF, while no survival difference was noted [39]. As a result, CMF is a good option for older patients with TNBC.

## 4. Conclusions

In summary, we reported a case of MGACA that was unique in more than one way. To our knowledge, neither a mixed-type carcinoma nor a carcinoma with apocrine differentiation has been previously associated with MGA. Moreover, this was the second case of a HER-2 neu positive MGACA. Thus, we expanded the spectrum of carcinomas associated with MGA, highlighting its heterogeneous nature, and suggesting the possibility of a simultaneous progression to triple-negative and HER-2neu positive carcinoma. Due to the rarity of MGA, more cases involving molecular studies are needed to better understand this interesting entity.

## Figures and Tables

**Figure 1 diagnostics-12-01458-f001:**
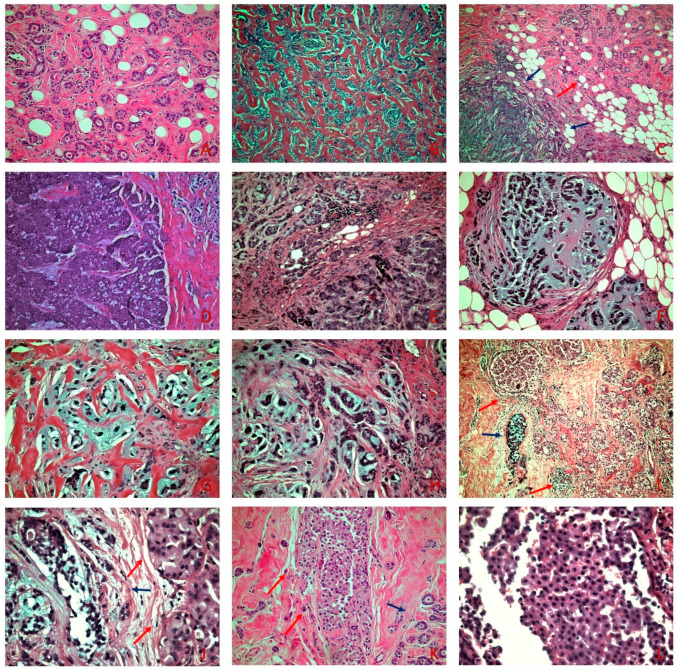
(**A**) MGA with small round uniform tubules lined with a single layer of cuboidal epithelial cells with occasional eosinophilic luminal secretion (H&E × 100). (**B**) AMGA displays areas of greater architectural complexity compared to MGA (H&E × 100). (**C**) MMPC (blue arrows) adjacent to MGA (red arrow) (H&E × 100) (**D**) Areas of solid invasive carcinoma adjacent to areas consisting of small nests with an abrupt transition to chondromyxoid matrix (H&E × 100). (**E**–**H**) Small nests, cords, single cells, tubular-like structures, and ring-like structures, embedded in a chondromyxoid matrix (H&E × 200). (**I**) On low power examination, PLCIS (red arrows) is adjacent to MMPC (blue arrow) (H&E × 40). (**J**) PLCIS (red arrows) shows cells with moderate to severe atypia, eosinophilic granular cytoplasm, and central to eccentric hyperchromatic nuclei adjacent to MMPC (blue arrow) (H&E × 200) (**K**) PLCIS (red arrows) adjacent to MGA (blue arrow) (H&E × 100). (**L**) On high power examination, lack of cellular cohesion is evident in the PLCIS (H&E × 400).

**Figure 2 diagnostics-12-01458-f002:**
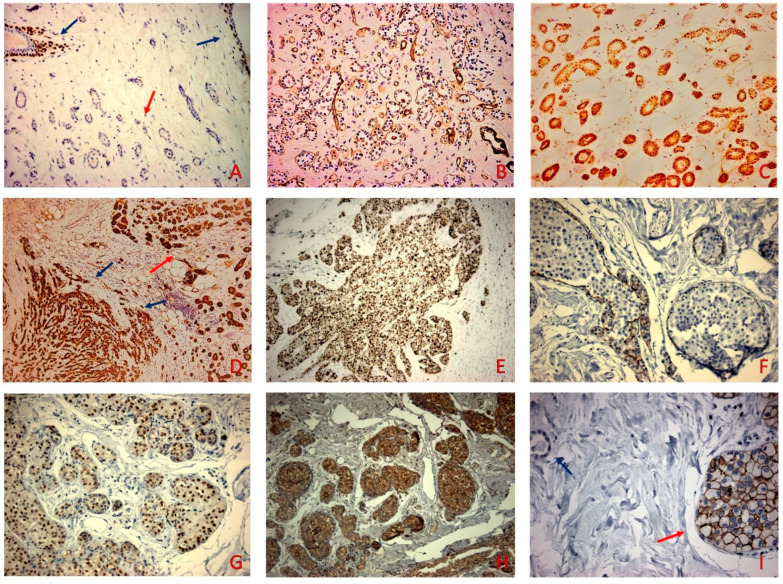
(**A**) P63 was negative in the MGA (red arrow) while retaining positive nuclear staining in the neighboring normal ducts and acini (blue arrows) (p63 × 100). (**B**) MGA was surrounded by Laminin (Laminin × 100). (**C**) S-100 was positive in the MGA (S-100 × 200). (**D**) S-100 positive staining in the MMPC (blue arrows) and the adjacent MGA (red arrow) (S-100 × 100). (**E**) Proliferation index Ki67 stained 70% of tumor cells (**F**) E-cadherin did not stain the cells of the intraductal proliferation (E-cadherin × 200) (**G**) AR stained the PLCIS (red arrow) and was negative in the MMPC (AR × 200). (**H**) PLCIS stained for P504S (P504S × 100). (**I**) PLCIS (blue arrow) showed 3+ staining and was negative in the MGA component (red arrow) (HER-2 × 400).

**Table 1 diagnostics-12-01458-t001:** Immunohistochemical stains performed in the four tumor components.

	MGA	AMGA	MMPC	Apocrine PLCIS
P63	−	−	−	−
CK5/6	−	−	−	−
SMA	−	−	−	−
Calponin	−	−	−	−
Laminin	+	+	−	−
S100	+	+	+	−
AR	−	−	−	+
AMACR	−	−	−	+
ER	−	−	−	−
PR	−	−	−	−
HER-2 neu	0	0	0	3+
Ki67	5%	10%	70%	10%
E-cadherin	+	+	+	−

Abbreviations: MGA; microglandular adenosis, AMGA; atypical microglandular adenosis, MMPC; metaplastic matrix-producing carcinoma, PLCIS; pleomorphic lobular carcinoma in situ.

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
