# Peer review of "Metaplastic Matrix-Producing Carcinoma and Apocrine Lobular Carcinoma In Situ Associated with Microglandular Adenosis: A Unique Case Report"

_diagnostics, 2022, doi:10.3390/diagnostics12061458_

Round 1

Reviewer 1 Report

The authors presented an unique and rare case of micro glandular adenosine associated carcinomas in the breast.

There are some suggestion:

  • Would you please elaborate the MRI results to gain more awareness for a small breast lession with contrast enhancement.
  • Please explain more detail in the introduction and discussion about the need to use panels of IHC to differentiate MGA, AMGA, MMPC, and apocrine PLCIS.
  • In a 79 year old patient with Stage I breast cancer of triple negative, please discuss the benefit and adverse effects of adjuvant chemotherapy. How about the international guideline, and explain why the authors decided to provide adjuvant chemotherapy CMF to this patient 

Author Response

We are grateful to the reviewers for highlighting the strengths and some weaknesses of the manuscript that we revised according to their comments. Below is a point-by-point answer to the comments and a list of the modifications we made in the manuscript.

Answer to Reviewer 1.

Would you please elaborate the MRI results to gain more awareness for a small breast lesion with contrast enhancement.

We elaborated the MRI results by adding a sentence as required by the reviewer.

Please explain more detail in the introduction and discussion about the need to use panels of IHC to differentiate MGA, AMGA, MMPC, and apocrine PLCIS.

We added a paragraph in the discussion section explaining the need for immunohistochemical and histochemical panels to differentiate these entities.

In a 79 year old patient with Stage I breast cancer of triple negative, please discuss the benefit and adverse effects of adjuvant chemotherapy. How about the international guideline, and explain why the authors decided to provide adjuvant chemotherapy CMF to this patient. 

We have added a couple of paragraphs to discuss the benefit and adverse effects of adjuvant chemotherapy. We have also explained the decision of the multidisciplinary tumor board to administer CMF adjuvant therapy according to international guidelines.

Except for the corrections and additions according to the reviewers' comments we also made a clarification in the case report section. We specify that P63 was positive peripherally in the myoepithelial cell layer in the PLCIS.

We would also like to ask the reviewers to allow us to change the title of the manuscript to “Metaplastic Matrix-producing Carcinoma and Apocrine Lobular Carcinoma In Situ Associated with Microglandular Adenosis. A Unique Case Report”. We believe that this title describes more accurately the case presented in our manuscript in contrast to the current one, which is more general.

Reviewer 2 Report

This manuscript by Koufopoulos and coworkers deals with a unique case report of microglandular adenosis associated with metaplastic matrix-producing carcinoma and HER-2 neu oncoprotein positive pleomorphic lobular carcinoma in situ with apocrine differentiation in a 79-year-old patient. In my opinion, this case report can be accepted after consideration of the following issues:

  1. The intro section describing Microglandular adenosis (MGA) is too short. The authors also stated that (The invasive carcinomas associated 46 with MGA belong to several different histologic subtypes). More details about some of these histologic subtypes should be mentioned. I recommend adding one or two paragraphs to the introduction.
  2. The authors stated (We present a unique case of MGA…). The authors did not mention if there is a similar case has been reported before or not? It is not clear if this is the first case of its kind to be reported. More explanation is needed.
  3. I recommend the manuscript should undergo an extensive English editing service. Typos errors should be checked.  
  4. Line 116, The number 1. Refers to what? 

Overall, I find the paper well written and the results clearly presented.

Author Response

We are grateful to the reviewers for highlighting the strengths and some weaknesses of the manuscript that we revised according to their comments. Below is a point-by-point answer to the comments and a list of the modifications we made in the manuscript.

Answer to Reviewer 2.

1. The intro section describing Microglandular adenosis (MGA) is too short. The authors also stated that (The invasive carcinomas associated with MGA belong to several different histologic subtypes). More details about some of these histologic subtypes should be mentioned. I recommend adding one or two paragraphs to the introduction.

1. The type of carcinomas associated with MGA are invasive carcinoma NST, MMPC, secretory carcinoma, carcinoma with metaplastic squamous differentiation, acinic cell carcinoma, spindle cell carcinoma, and adenoid cystic carcinoma. We have added this information to the introduction section as the reviewer has requested us to.

2. The authors stated (We present a unique case of MGA…). The authors did not mention if there is a similar case has been reported before or not? It is not clear if this is the first case of its kind to be reported. More explanation is needed.

2. There is no similar case in the English literature of the co-existence of two types of carcinoma associated with microglandular adenosis. Also, microglandular adenosis has been associated with several subtypes of triple negative breast carcinoma, rarely with cases of Her-2 positive breast carcinoma but never with a combination of a triple negative and Her-2 positive breast carcinoma. We explain this in the discussion section (lines 166 to 171)

3. I recommend the manuscript should undergo an extensive English editing service. Typos errors should be checked.  

3. The manuscript underwent extensive English editing as required by the reviewer.

4. Line 116, the number 1. Refers to what? 

4. Number 1 in line 116 should be moved in line 115 indicating “Table 1”.

We also made a clarification in the case report section. We specify that P63 was positive peripherally in the myoepithelial cell layer in the PLCIS.

We would also like to ask the reviewers to allow us to change the title of the manuscript to “Metaplastic Matrix-producing Carcinoma and Apocrine Lobular Carcinoma In Situ Associated with Microglandular Adenosis. A Unique Case Report”. We believe that this title describes more accurately the case presented in our manuscript in contrast to the current one, which is more general.

Round 2

Reviewer 2 Report

I found no problem accepting this case report